# Synergistic Antibacterial Effect of Zinc Oxide Nanoparticles and Polymorphonuclear Neutrophils

**DOI:** 10.3390/jfb13020035

**Published:** 2022-03-23

**Authors:** Kai Ye, Moran Huang, Xiaojian He, Zhiquan An, Hui Qin

**Affiliations:** 1Department of Orthopedic Surgery, Shanghai Jiao Tong University Affiliated Sixth People’s Hospital, NO.600 Yishan Road, Shanghai 200233, China; yekai706@163.com (K.Y.); huangmoran44@126.com (M.H.); 2Department of Orthopaedics, Affiliated Anhui Provincial Hospital of Anhui Medical University, NO.17 Lujiang Road, Hefei 230001, China; 3Department of Orthopaedics, Qingpu Branch of Zhongshan Hospital, Fudan University, NO.1158 Gongyuandong Road, Shanghai 201700, China; hexiaojian1@126.com

**Keywords:** antimicrobial property, polymorphonuclear leukocyte, synergistic effect, zinc oxide nanoparticle

## Abstract

Zinc oxide nanoparticles (ZnONPs) are inorganic nano-biomaterials with excellent antimicrobial properties. However, their effects on the anti-infection ability of the innate immune system remains poorly understood. The aim of the present study was to explore the potential immunomodulatory effects of ZnONPs on the innate immune system, represented by polymorphonuclear leukocytes (PMNs), and determine whether they can act synergistically to resist pathogen infections. In vitro experiment showed that ZnONPs not only exhibit obvious antibacterial activity at biocompatible concentrations but also enhance the antibacterial property of PMNs. In vivo experiments demonstrated the antibacterial effect of ZnONPs, accompanied by more infiltration of subcutaneous immune cells. Further ex vivo and in vitro experiments revealed that ZnONPs enhanced the migration of PMNs, promoted their bacterial phagocytosis efficiency, proinflammatory cytokine (TNF-α, IL-1β, and IL-6) expression, and reactive oxygen species (ROS) production. In summary, this study revealed potential synergistic effects of ZnONPs on PMNs to resist pathogen infection and the underlying mechanisms. The findings suggest that attempts should be made to fabricate and apply biomaterials in order to maximize their synergy with the innate immune system, thus promoting the host’s resistance to pathogen invasion.

## 1. Introduction

Pathogenic bacterial infections in the orthopedic field, including implant and periprosthetic joint infections, constitute common clinical challenges that require repeated surgery and time-consuming therapy [1]. The incidence of orthopedic device infections varies from 2% to 5% clinically [2], and the dominating pathogens are *Staphylococcus* species, particularly *Staphylococcus aureus* (*S. aureus*) [3]. Devastating outcomes along with unacceptable economic burden are usually associated with implant and periprosthetic joint infections [4].

The immune system plays a crucial role in controlling pathogenic infections [5,6], as it is the body’s main line of defense against pathogen invasions. Specifically, polymorphonuclear leukocytes (PMNs or neutrophils) are among the cell types that are mobilized as a part of the host’s innate immune response in the initial stages, toward the site of infection [7]. PMNs can effectively neutralize pathogens through various mechanisms, including phagocytosis [8], chemotaxis [9], and superoxide production [10], thereby effectively preventing the development of infections. Hence, a fully functional PMN response represents a fundamental component of the host immune response to invading pathogens.

Recently, the emergence of nanoparticle-based antimicrobial agents has provided a new paradigm for antimicrobial treatments. These agents, including gold [11], silver [12], and copper [13] nanoparticles, possess good antibacterial properties in vitro. However, they are often criticized for their lack of corresponding effects in vivo [14]. Several studies have demonstrated limited in vivo antibacterial effects of implants coated with antimicrobial nanoparticles; thus, redundant efforts have been made for complicated modifications that are associated with high cost and erratic yield [15,16]. During bacterial infections, the phagocytic immune response can be suppressed by virulence factors liberated by bacteria [17,18]. Bacterial biofilms on implants can also impair microbial eradication by inhibiting phagocytosis of immune cells [19]. However, the present practice of nanoparticle-based antimicrobial treatment invariably neglects the restoration of immunosuppression, which may explain the inability of nanoparticles to maximize their antibacterial effects in vivo. Knowledge of interaction between nanoparticles and the immune system will help reduce in vivo applications of these agents. In other words, the development of agents that can mobilize or enhance the antibacterial ability of the body’s immune system and simultaneously have superior antibacterial properties might be a breakthrough in the anti-infection research of nanomaterials [20,21,22].

Zinc oxide nanoparticles (ZnONPs) constitute a common type of nanoparticle with excellent antimicrobial properties [23,24]. They have the ability to mobilize immunoreaction. ZnONPs reportedly promote the proinflammatory response by activating monocytes [25] and releasing interleukin-8 by human bronchial epithelial cells [26]. There is also evidence that ZnONPs enhance the ability of PMNs to exert phagocytosis and degranulation [27,28,29]. The nanocomposites of ZnO/Ag nanoparticles armed in chitosan/polyethylene oxide nanofibers have been successfully applied to exert synergic antibacterial effect with fibroblast cells [30]. Nonetheless, the actual effect of the interaction between ZnONPs and the innate immune system in the context of protecting the host from pathogenic infection remains unclear [31,32,33].

Considering that the neutrophils are the first-line responders of the host’s innate immune system [5,6,7], the exploration of the antibacterial effect under the interaction between ZnONPs and neutrophils may be a paradigm for the future application of nanomaterials that act synergistically with the host immune system to eliminate bacteria in early infection stages. In this study, in vitro nanoparticle–PMN–bacterial co-culture and in vivo *S.*
*aureus*-infected air pouch models were utilized to explore the potential effects and underlying mechanisms of ZnONPs on neutrophils to resist pathogen infection.

## 2. Materials and Methods

### 2.1. Materials

#### 2.1.1. Characterization of ZnONP Nanoparticles

ZnO nanoparticle dispersions with an average particle size of <20 nm were purchased from Maogon Nano (Shanghai, China). Particle morphology and size were characterized using TEM (JSM-IT200; JEOL, Tokyo, Japan). The chemical constituents and states of the elements were characterized by XPS (Escalab 250Xi; Thermo Fisher Scientific, Waltham, MA, USA).

#### 2.1.2. Human PMN Isolation

This study was conducted in accordance with the principles of the Declaration of Helsinki, and ethical approval was granted by the review board of our institution. Heparinized venous blood was obtained from healthy volunteers after informed consent was obtained. PMNs were isolated by the neutrophil separating Kit (Solarbio, Beijing, China) using the gradient density centrifugation method [34]. Briefly, 4 mL of Solution A, 2 mL of Solution C, and 4 mL of heparinized human venous blood were successively added into a 15 mL centrifuge tube. Then, the tube was centrifuged at 610× *g* for 30 min with the lowest accelerated and retarded speeds. The neutrophils were carefully removed into another 15 mL centrifuge tube, resuspended with 5 mL Red Blood Cell Lysis Buffer, reacted for 5 min, and centrifuged at 250× *g* for 10 min. After discarding the supernatant, the neutrophils were then resuspended with 5 mL of PBS (Ca^2+^/Mg^2+^ free) and centrifuged at 250× *g* for 10 min. The remaining neutrophils were finally resuspended with RPMI-1640 culture medium. Cell viability was determined by trypan blue exclusion method.

#### 2.1.3. Bacterial Preparation

Freeze-dried gram-positive *S. aureus* (ATCC 43300) was acquired from the American Type Culture Collection (Manassas, VA, USA). Before each experiment, the bacteria were inoculated onto sheep blood agar plates and grown overnight at 37 °C. An individual colony was then selected, transferred into 4 mL of fresh trypticase soy broth (TSB) medium, and incubated on an orbital shaker overnight at 37 °C. The bacterial density was estimated by optical density (OD) at a wavelength of 600 nm and was finally maintained at 0.8–1.0. The bacteria were gradually diluted for subsequent experiments.

### 2.2. Biocompatibility Evaluation

Using a 96-well plate, PMN (1 × 10^5^ per well) suspensions were incubated in RPMI-1640 complete culture medium (Hyclone Laboratories Inc., Logan, UT, USA) with ZnONPs included at various concentrations (0, 25, 50, 100, and 200 μg mL^−1^) at 37 °C in a humidified atmosphere containing 5% CO_2_. After co-culturing for 4 h, the medium was substituted with 0.1 mL RPMI-1640 culture medium. Next, 10 μL of CCK-8 solution (Dojindo, Kumamoto, Japan) was added to each well and incubated at 37 °C for 1 h. Finally, the plate was assessed with a microplate reader (Thermo Fisher Scientific, Waltham, USA) at a wavelength of 450 nm. An LDH leakage assay was also performed using an LDH assay kit (Beyotime, Shanghai, China), according to the manufacturer’s protocol.

### 2.3. In Vitro Antibacterial Evaluation

To assess the antibacterial activity of ZnONPs, 1 mL of bacterial suspension (1 × 10^5^ CFU mL^−1^) was incubated with ZnONPs (0, 25, 50, and 100 μg mL^−1^) in 24-well plates at 37 °C for 4 h. The SPM was then used to evaluate the bacterial count of the culture medium with different nanoparticle concentrations. To further assess the antibacterial activity of PMNs under the influence of ZnONPs, 1-mL PMN suspensions (1 × 105 cells mL^−1^) were incubated with bacteria (1 × 10^5^ CFU mL^−1^) and ZnONPs (0, 25, 50, and 100 μg mL^−1^) in 24-well plates at 37 °C for 4 h. SPM was also used to estimate the bacterial count, which was presented as log_10_ CFU.

### 2.4. In Vivo S. aureus-Infected Air Pouch Model

#### 2.4.1. In Vivo *S. aureus* Infected-Air Pouch Model

The animal operations were approved by our Animal Care and Experiment Committee and performed in accordance with the NIH Guide for Care and Use of Laboratory Animals guidelines. Twenty-four 8-week-old female Sprague–Dawley (SD) rats were acclimated for 1 week under standard conditions. Before each operation, the rats were anesthetized intraperitoneally with 3% pentobarbital sodium. The operation area was located at the intra-scapular area, which was shaved and sterilized twice with 70% ethanol. Using a 20 mL syringe and a 0.2 μm sterile filter, 20 mL of sterile air was injected subcutaneously to generate an air pouch. Three days later, the rats were anesthetized again, and an additional 10 mL of sterile air was injected to maintain the air pouch. Six days after the first injection, the rats were randomly divided into four groups (*n* = 6): control group, *S. aureus*-infected group, 50 μg mL^−1^ ZnONPs + *S. aureus*-infected group, and 100 μg mL^−1^ ZnONPs + *S. aureus*-infected group. Next, 2 mL of PBS, *S. aureus* (1 × 10^7^ CFU mL^−1^) with ZnONPs (50 μg mL^−1^), and *S. aureus* (1 × 10^7^ CFU mL^−1^) with ZnONPs (100 μg mL^−1^) were injected into the air pouches of each group member.

#### 2.4.2. In Vivo Microbiological Assessment

The exudate and the tissue of the air pouch was harvested 24 h after the final injection. Briefly, before six rats from each group were euthanized, 5 mL of sterile PBS was injected into the air pouch, and the contents were mixed by massaging them gently. A sagittal incision was made to collect the total volume of the exudate. The number of bacteria was calculated using the SPM.

#### 2.4.3. Histological Study

Three rats from each group were euthanized, and partial full-thickness skin within the air pouch area was harvested and fixed with 10% paraformaldehyde solution at room temperature. After being washed with fresh PBS, the skin samples were dehydrated with an alcohol gradient and embedded in paraffin. Histological sections were then prepared using a sledge microtome (Leica, Hamburg, Germany). Finally, sample sections were deparaffinized with a xylene solution, subjected to H&E and Giemsa staining, and observed under an optical microscope (Nikon, Tokyo, Japan).

### 2.5. In Vitro Mechanism Study

#### 2.5.1. Chemotactic Migration of PMNs

For the chemotactic migration assay, RPMI-1640 complete medium loaded with 1 × 10^5^ PMNs and various concentrations of ZnONPs (0, 25, 50, and 100 μg mL^−1^) in a volume of 0.2 mL were added into the upper chambers of 24-well transwell plates (pore size: 3 μm; Corning, NY, USA). The lower chamber was loaded with 0.5 mL of RPMI-1640 complete medium containing various concentrations of ZnONPs (0, 25, 50, and 100 μg mL^−1^) and bacterial suspensions (1 × 10^6^ CFU mL^−1^). After being incubated for 2 h at 37 °C, the non-migrating PMNs were carefully removed using a swab. The migrated PMNs in the membrane were dipped in 4% paraformaldehyde for 10 min, transferred into 0.1% crystal violet for 15 min and gently washed thrice with PBS. The chambers were viewed under an optical microscope (Nikon, Tokyo, Japan).

#### 2.5.2. Quantitative Real-Time PCR Detection of Proinflammatory Cytokines Released by PMNs In Vitro

To evaluate the expression of tumor necrosis factor (TNF)-α), interleukin (IL)-1β, and IL-6, 1 mL of 1 × 10^6^ PMNs was incubated with various concentrations of ZnONPs (0, 25, 50, and 100 μg mL^−1^) and a bacterial suspension (1 × 10^7^ CFU mL^−1^) in 24-well plates at 37 °C for 4 h. The culture medium was then centrifuged at 250× *g* for 10 min and resuspended in TRIzol reagent (Invitrogen, Waltham, MA, USA). Total RNA from PMNs was extracted and quantified according to the manufacturer’s protocol. Total RNA (1 μg) was reverse transcribed into cDNA using the PrimeScript RT reagent Kit (TaKaRa, Shiga, Japan). RT-qPCR was performed using SYBR Premix Ex Taq (Takara) with a LightCycler480 system (Roche, Mannheim, Germany).

#### 2.5.3. Detection of Phagocytosis of PMNs Isolated from Human Whole Blood Using a Bacteria–Blood–ZnONPs Co-Culture Model

Whole blood (0.5 mL) was mixed with bacteria (1 × 10^7^ CFU mL^−1^) and ZnONPs (0, 25, 50, and 100 μg mL^−1^), and subsequently incubated in a 24-well plate at 37 °C for 0, 5, 15, 30, 45, 60, and 90 min. At each time point, three random sample smears for each group were created for Wright–Giemsa staining and observed under an optical microscope (Nikon, Tokyo, Japan). The phagocytosis rate was calculated as follows. Six isolated microscopic fields were evaluated for each group at each time point. The number of phagocytosed, adherent, and dissociated bacteria within 10 μm of a PMN was counted. Phagocytosis rate of PMN (%) = (phagocytosed count/the sum of phagocytosed, bound, and dissociated bacteria count) × 100.

#### 2.5.4. Cell Lysis of PMNs during the Antibacterial Process

A 10-μL suspension of bacteria (1 × 10^7^ CFU mL^−1^) was added to 0.1 mL of PMNs (1 × 10^5^ cells mL^−1^) and co-cultured with various concentrations of ZnONPs (0, 25, 50, and 100 μg mL^−1^) in 96-well plates at 37 °C for 1.5 h. LDH release was detected by an LDH assay kit (Beyotime, Shanghai, China). The evaluation was based on the OD value at a wavelength of 490 nm. Cell lysis rate (%) = (experimental group–control group)/(cell disruption group–control group).

#### 2.5.5. Reactive Oxygen Species (ROS) Production by PMNs In Vitro

ROS production was measured by a ROS assay kit (Beyotime, Shanghai, China). To evaluate the influence of ZnONPs on the capacity of PMNs to produce ROS, 0.1 mL of the latter (1 × 10^5^ mL^−1^ cells mL^−1^) was co-cultured with ZnONPs (0, 25, 50, and 100 μg mL^−1^) in 96-well plates at 37 °C for 1.5 h. To evaluate the effect of ZnONPs on the capacity of PMNs to produce ROS during the antibacterial process, 10 μL of a bacterial suspension (1 × 10^7^ CFU mL^−1^) was added simultaneously. Next, 2′,7′-dichlorodihydrofluorescein diacetate (DCFH-DA) diluted with serum-free RPMI-1640 medium was added into each well at a final concentration of 10 μM. After an additional incubation time of 30 min in the dark, the cells were centrifuged and washed twice with PBS to remove the residual DCFH-DA. Fluorescence intensity was examined under an inverted fluorescence microscope, and quantitative measurements were conducted using a fluorescence microplate reader (Thermo Fisher Scientific, Waltham, MA, USA).

#### 2.5.6. Superoxide Production by PMNs In Vitro

Following the manufacturer’s protocol for the Superoxide Assay Kit (Beyotime, Shanghai, China), PMNs were collected, centrifuged, and resuspended in a 0.2 mL working solution at a final concentration of 1 × 10^5^ cells mL^−1^. The PMNs were then incubated with various concentrations of ZnONPs (0, 25, 50, and 100 μg mL^−1^) in 96-well plates, with or without bacteria (1 × 10^6^ CFU mL^−1^), for 2 h. The OD value was measured at a wavelength of 450 nm.

#### 2.5.7. Statistical Analysis

Experimental data were displayed as the mean ± standard deviation. Statistical analyses were conducted by one-way analysis of variance and two-tailed Student’s *t*-test. SPSS software (version 25.0, IBM Corp, Armonk, NY, USA) was used to perform statistical analyses. Significance was set at a *p* value < 0.05.

## 3. Results

### 3.1. Characterization of ZnONPs

The transmission electron microscopy (TEM) results showed that the nanostructured ZnONPs had a particle size of approximately 20 nm (Figure 1A). The elemental composition of the ZnONPs was further determined by X-ray photoelectron spectroscopy (XPS; Figure 1B,C). Results showed that the binding energies of Zn2p were at 1044.6 eV for Zn2p1 and 1021.5 eV for Zn2p3, indicative of the characteristic Zn signals.

### 3.2. In Vitro Biocompatibility Evaluation of ZnONPs

PMN cell viability was detected following incubation with ZnONPs, using the CCK-8 assay (Figure 2A). After 4 h of incubation, ZnONPs at concentrations of 25, 50, and 100 μg mL^−1^ did not significantly affect the viability of PMNs compared with the control group, whereas 200 µg mL^−1^ resulted in significantly decreased viability. This was further verified by a lactate dehydrogenase (LDH) leakage assay (Figure 2B). LDH release was equivalent between the control group and the groups treated with ZnONPs at 25, 50, and 100 μg mL^−1^; however, it was increased significantly in the group treated with 200 μg mL^−1^ ZnONP. Therefore, concentrations of 25, 50, and 100 μg mL^−1^ were used in subsequent experiments.

### 3.3. In Vitro Antimicrobial Evaluation of ZnONPs and PMNs

Antibacterial activity was verified using the spread-plate method (SPM) and demonstrated using log_10_ CFU mL^−1^ values. After 4 h of incubation, the ZnONP-treated groups showed obvious antibacterial activity compared with the positive control group (Figure 3A,B). This effect was more obvious in the group treated with a ZnONP concentration of 100 μg mL^−1^ than in the other two groups (25 and 50 μg mL^−1^). The bacterial count decreased in the PMN-treated group. Interestingly, the bacterial counts decreased further in the PMN-treated groups under the influence of ZnONPs at various concentrations and this decrease presented in a dose-dependent manner.

### 3.4. In Vivo Antibacterial Evaluation of ZnONPs Using the Infected Air Pouch Model

The bacterial count of the lavage harvested from the air pouch was evaluated using SPM. The bacterial count decreased in the ZnONP groups in a concentration-dependent manner compared to the positive control group (Figure 4A). This was further verified by quantitative analysis using log_10_ CFU mL^−1^ values (Figure 4B). These results indicate that ZnONPs exhibit antibacterial activity in vivo.

### 3.5. Histological Evaluation of the S. aureus-Infected Air Pouch Model

Hematoxylin and eosin (H&E) staining, and Giemsa staining, were performed for the histological examination of the harvested air pouch (Figure 5). In the H&E stained slices, immune cell infiltration was the lowest in the control group, while the an obvious immune cell infiltration was observed in the bacteria-infected group; this effect was promoted by ZnONPs treatment, in a concentration-dependent manner. Moreover, the Giemsa staining results also reflected the antibacterial function of ZnONPs, with more bacteria observed in the bacteria-infected group, fewer observed in the ZnONP-treated groups, and no bacteria detected in the control group.

### 3.6. ZnONP Treatment Enhances PMN Migration and Cytokine Expression In Vitro

The transwell assay facilitated the detection of PMN migration activity under the influence of ZnONP nanoparticles. The number of PMNs that migrated through the membrane increased in following treatment with ZnONP, in a concentration-dependent manner (Figure 6A). The quantitative cell count in the transwell test also reflected this result (Figure 6B). Further, the ZnONP, PMN, and bacteria co-culture increased the mRNA expression of inflammation-related cytokines (TNF-α, IL-1β, and IL-6) in a dose-dependent manner (Figure 6C–E). These results suggest that ZnONPs promote PMN migration and proinflammatory cytokine expression.

### 3.7. ZnONPs Increase Phagocytosis of Bacteria by PMNs in Human Whole Blood

To further explore the effects of ZnONPs on PMNs, a phagocytosis assay was conducted with human whole blood (Figure 7A). In the established bacteria–blood–ZnONP co-culture model, the number of bacteria ingested by PMNs increased gradually with incubation time. Meanwhile, the number of bacteria phagocytosed by PMNs was significantly increased in co-cultures with ZnONPs.

The mean phagocytosis rates of the untreated group and ZnONPs treated groups (0, 25, 50, and 100 μg mL^−1^) were 8%, 10%, 10%, and 11%, respectively, at 5 min; 25%, 26%, 30%, and 33%, respectively, at 15 min; 40%, 50%, 53%, and 62%, respectively, at 30 min; 47%, 61%, 72%, and 80%, respectively, at 45 min; 54%, 65%, 77%, and 88%, respectively, at 60 min; and 55%, 66%, 78%, and 89%, respectively, at 90 min. The overall trend of the phagocytosis rate was the lowest in the untreated group and higher in the ZnONP groups, in a concentration-dependent manner, with a more significant effect observed after 30 min of incubation (Figure 7B), indicating that ZnONPs demonstrated a synergistic immune regulatory function with PMNs by promoting phagocytosis.

### 3.8. ZnONPs Cause Slight Decrease in Lysis of PMNs Isolated from Human Whole Blood

The degree of PMN lysis was also determined with the average PMN lysis rate found to be 14.6%, 9.3%, 12.5%, and 12.1% in groups treated with ZnONP concentrations of 0, 25, 50, and 100 μg mL^−1^, respectively (Figure 7C). Thus, ZnONPs slightly, yet significantly, attenuated the degree of PMN lysis.

### 3.9. ZnONPs Activate PMN ROS and Superoxide Production In Vitro

To further investigate the enhanced antibacterial effect of PMNs in the presence of ZnONPs, as well as the mechanism of their synergistic effects on the immune system, ROS and superoxide production were detected. Fluorescence levels were negligible in the untreated group, with a slight increase observed following incubation of PMNs with ZnONPs (Figure 8A). Co-culturing with bacteria also induced an increase in ROS production by PMNs; however, this effect was amplified in the presence of ZnONPs, in a concentration-dependent manner. These results were confirmed by quantitative measurement of ROS fluorescence (Figure 8B). Moreover, superoxide production exhibited a similar tendency to that observed for ROS production (Figure 8C). These results suggest that the enhanced antibacterial effect elicited by PMNs, following co-culture with ZnONPs, may be due to increased production of ROS and superoxide.

## 4. Discussion

In this study, we found that during bacterial infection, ZnONPs enhance the antibacterial property of PMNs, alleviate subcutaneous immune cell infiltration in vivo, enhance PMNs migration, improve bacterial phagocytosis efficiency of PMNs, promote PMNs to express the proinflammatory cytokines (TNF-α, IL-1β, and IL-6), and facilitate reactive oxygen species (ROS) production of PMNs. ZnONPs reportedly possess excellent antimicrobial properties, and they have been applied as antibacterial agents to prevent indwelling device infections [35,36]. Nanoparticles of various particle sizes, generally from 5 to 300 nm, have been reported to possess sufficient antimicrobial properties [37,38]. The antibacterial activity of ZnONPs is size dependent—ZnONPs of smaller particle sizes can easily penetrate bacterial membranes because of the large interfacial area [39]. However, decreased particle size inevitably affects biocompatibility [40]. The mechanism of cytotoxicity attributed to the dissolution of ZnONPs and the generation of ROS [20]. In this study, the particle size of ZnONPs was 20 nm, and these particles showed a good antibacterial efficacy in vitro. The chosen particle size is for the balance between antimicrobial properties and cell viability. Indeed, previous studies have confirmed that 20-nm ZnONPs at adequate concentrations possessed good biocompatibility [21]. The antimicrobial properties and biocompatibility are also influenced by nanoparticle concentration. Thus, we verified the cytocompatibility of ZnONPs and found no distinct cytotoxicity at a concentration gradient of below 100 μg mL^−1^. This result is consistent with the findings of previous studies, which set 100 μg mL^−1^ as the default biosafety concentration for PMNs [27,28,29]. Thus, the antibacterial effect and biocompatibility determined in this study equilibrated at the chosen concentrations.

Neutrophils are among the cell types that are mobilized as a part of the host’s innate immune response in the early stages; they act as first-line responders against pathogenic invasion [5,6,7]. Here, we found that the combination of ZnONPs and PMNs further reduced bacterial load in vitro. An in vitro study showed that ZnONPs enhance the phagocytosis of human neutrophils [27]. Another study reported that ZnONPs positively modulate the degranulation process in human neutrophils [28]. Several studies have also demonstrated the pro-inflammatory effect of ZnONPs on the innate immune system including macrophages [21,41]. Given the positive immunoregulation and pro-inflammatory effect of ZnONPs on neutrophils, they may exert a synergistic antibacterial effect, rather than exerting the effect independently. To further manifest the antibacterial effect of ZnONPs in vivo, we applied an *S. aureus*-infected air pouch model—a typical model to determine the antibacterial property of certain agents and reflect acute inflammation [42]. The results showed that ZnONPs significantly reduced bacterial count in the harvested exudate; Giemsa staining also showed a decrease in bacterial proportion in the subcutaneous tissue within the air pouch. These results were consistent with those of in vitro experiments. A previous study has also demonstrated that ZnONPs significantly attenuated the severity of infection in a localized soft tissue infection model owing to their antimicrobial property [43]. Interestingly, even though bacterial burden was relieved under the intervention of ZnONPs, the subcutaneous immune cell infiltration was increased following ZnONP intervention, which indicated a potential promotion effect of ZnONPs on the innate immune system. It is worth mentioning that although local bacterial burden was relieved by ZnONPs, substances released by multiple bacteria may still trigger immune cell migration. Nevertheless, the exact effect of ZnONPs on neutrophils during antibacterial process needs further verification.

When encountering local pathogen infection, the host exerts inflammatory responses by recruiting multiple immune cells from the blood, nearly 70% of which are neutrophils—the first leukocytes to be recruited to an infected site [7,44,45]. Excessive mobilization and recruitment of neutrophils are a prerequisite for local tissue cell infiltration [44,46]. The results of the Transwell assay in our study showed that ZnONPs significantly facilitated neutrophil recruitment. This effect was possibly mediated by the release of zinc ions that dissolved from ZnONPs in aqueous medium [47,48]. During the neutrophil recruitment cascade, neutrophils in the circulation are mobilized by the change in the surface of endothelium. This process results from the stimulation of inflammatory mediators, including cytokines that are released from tissue-resident neutrophils in contact with bacteria [7]. Using the ZnONP, PMN, and bacterial co-culture system, we proved that ZnONPs enhanced proinflammatory cytokine expression in PMNs at the mRNA level. Although a few studies have reported the proinflammatory effect of ZnONPs on PMNs during antibacterial process, several studies have demonstrated proinflammatory properties of ZnONPs on other types of immune cells. Poon et al. [25] proved that ZnONPs enhanced TNF-α and IL-1β expression in THP-1 (acute monocytic human leukemia) cells. ZnONPs were also found to stimulate macrophages and primary dendritic cells to release TNF-α and IL-6 [21]. Roy et al. [41] demonstrated that ZnONP-activated macrophages upregulated TNF-α, IL-1β, and IL-6 expression depending on MAPK signaling. Excessive production of proinflammatory cytokines could further trigger consecutive antibacterial actions of neutrophils including immune cell recruitment, phagocytosis, and ROS production [49].

After reaching the inflammatory site and experiencing multiple stimuli, neutrophils are more activated as phagocytic cells than circulating neutrophils [46]. Multitudinous of stimuli within the inflammatory site, such as lipopolysaccharide, chemokines, and cytokines, will facilitate neutrophils to mount an increased activation following successive stimuli [50]. However, in the orthopedic field, conditions become more complex. This is because in early-stage infections after orthopedic surgery, bleeding around indwelling devices caused by surgical operation invariably constitutes a unique microenvironment. In this circumstance, the blood becomes a medium where interaction occurs between implants, neutrophils, and pathogens. To further evaluate the effect of ZnONPs on phagocytic behavior of neutrophils, and to best simulate in vivo conditions, phagocytosis, one of the core functions of neutrophils in eliminating pathogens [50], was evaluated using the ex vivo bacteria–blood–ZnONP co-culture model. The results demonstrated that ZnONPs enhanced the efficiency of PMNs to phagocytose bacteria. This promoting effect is in accordance with a previous study finding that ZnONPs increase the ability of neutrophils to phagocytose sheep red blood cells and fluorescent latex beads [27]. Indeed, an increased phagocytic capacity of macrophages promoted by zinc ions is a common finding both in vitro and in vivo [41,51,52]. In this study, neutrophils gathered within the inflammatory site were activated and behaved as phagocytic cells. Zinc ions released from ZnONPs may concentrate within the phagolysosomes of activated neutrophils to form a high zinc concentration environment, thus exerting an enhanced bactericidal efficiency through specific mechanisms, for example, the destruction of the Fe-S cluster [53,54]. It is worth mentioning that the rapid phagosomal maturation and phagocytosis within neutrophils are not perfect processes; thus, oxidative products and cytolytic contents may be released from the cells, damaging the surrounding tissues. Unexpectedly, we confirmed the protective effect of ZnONPs on PMNs via a slight decrease in cell lysis rate during antibacterial process. This cytoprotective effect might be explained by the results of Girard et al. [29], who demonstrated a delayed neutrophil apoptotic effect under the influence of ZnONPs through a de novo protein synthesis-dependent mechanism.

After the encapsulation of bacteria in phagosomes, one of the essential mechanisms of immune cells is to kill them in a nicotinamide adenine dinucleotide phosphate (NADPH) oxygenase-dependent manner [46], also referred to as the respiratory burst. During this process, ROS are a vital weapon for neutrophils as they contribute to the destruction of bacteria [10], whereas superoxide anions serve as the initiating switch leading to the generation of ROS [49]. In this study, neither ZnONPs nor bacterial cells alone induced significantly enhanced ROS or superoxide production by PMNs. In this context, previous studies have proved that ZnONPs at bio-safe concentrations will not directly enhance ROS production in neutrophils [29]. We further observed that ROS and superoxide anions were mass-produced under the presence of both ZnONPs and bacteria. Under such circumstances, ZnONPs are more likely to function as priming agents, which might trigger certain pathways of PMNs, thereby increasing ROS production. It is worth noting that NOX2 functions as the catalytic unit of NADPH oxidase, which is an essential protein in ROS and superoxide production. [55] Hence, we speculated that ZnONPs might improve ROS production through NOX2-related mechanisms. In addition, NF-κB-related genes also play a role in stimulating ROS production, [56] and ZnONPs can reportedly activate NF-κB signaling in THP-1 cells [57]. However, further studies are required to clarify the precise mechanism underlying the promotion of ROS production.

This study has proved the potential synergistic effects of ZnONPs on PMNs to resist pathogen infection. Further research should be devoted to reducing the toxicity of ZnONPs through various methods, including forming complexes with substances such as chitosan [30], or embedding on implant surfaces [43]. These attempts may be expected to increase the concentration of ZnONPs within the biosafety range, thus achieving a better bactericidal effect. On the other hand, how to further increase immune cell infiltration in the infected site is also an urgent problem. ZnONPs were reportedly produce massive ROS under ultraviolet irradiation [24]. A possible paradigm is to use UV light irradiation at specific times to induce local neutrophil apoptosis, which is expected to further trigger immune cell recruitment [33]. In addition, ZnONPs applied on the animal model of subcutaneous soft tissue infection were proved to be effective in this study. We believe that these results will be equally valid in other models including humans; however, further research is needed. This study had certain limitations. First, neutrophils are not the only immune cells in the innate immune system. Indeed, many cells, including macrophages, play a crucial role in anti-infection immunity [41,52]. Hence, focusing exclusively on neutrophil function could have resulted in a lack of comprehensiveness. Second, the antibacterial effect of PMNs is not limited to phagocytosis and ROS production. Other mechanisms including neutrophil extracellular traps [8] and degranulation [28] are of importance, and they were not explored in this study.

## 5. Conclusions

ZnONPs have potential synergistic effects on neutrophils to resist pathogen infection along with promoting neutrophil migration, proinflammatory cytokine mRNA expression, phagocytosis, and ROS production. The findings help to fabricate and apply biomaterials in order to maximize their synergy with the innate immune system, thus promoting the host’s resistance to pathogen invasion.

## Figures and Tables

**Figure 1 jfb-13-00035-f001:**
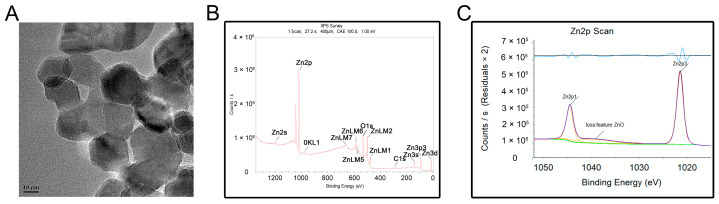
Material characterization of zinc oxide nanoparticles (ZnONPs). (**A**) Transmission electron microscopy of ZnONPs. (**B**,**C**) X-ray photoelectron spectroscopy showed the elementary composition of ZnONPs.

**Figure 2 jfb-13-00035-f002:**
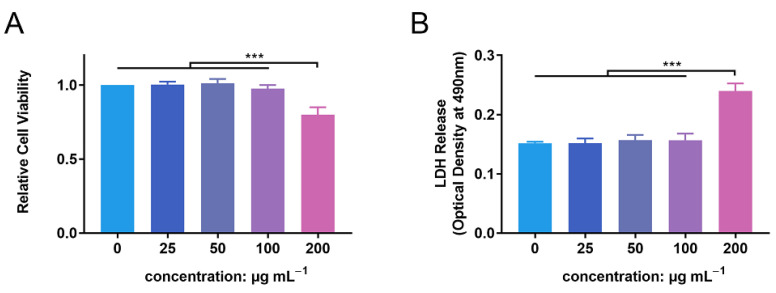
Cytotoxicity of zinc oxide nanoparticles (ZnONPs) on polymorphonuclear neutrophils (PMNs). (**A**) Relative cell viability of PMNs after co-culture with ZnONPs at various concentrations (0, 25, 50, 100, 200 μg mL^−1^) for 4 h using CCK-8 assay. (**B**) Lactate dehydrogenase (LDH) release of PMNs treated with ZnONPs at various concentrations (0, 25, 50, 100, 200 μg mL^−1^) for 4 h. *** *p* < 0.001.

**Figure 3 jfb-13-00035-f003:**
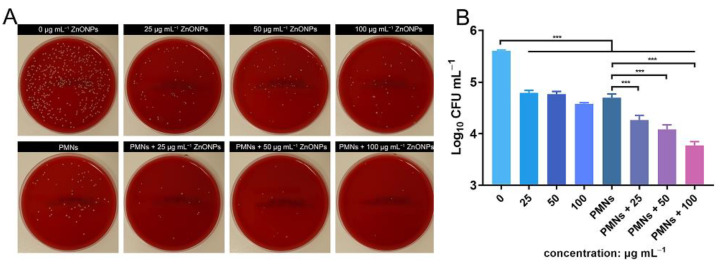
In vitro evaluation of the antibacterial activity elicited by zinc oxide nanoparticles (ZnONPs) alone and polymorphonuclear neutrophils (PMNs) treated with ZnONPs for 4 h. (**A**) Representative photos of the blood plate medium used for co-culturing of *Staphylococcus aureus* and ZnONPs (0, 25, 50, 100 μg mL^−1^), with or without PMNs. (**B**) Quantitative evaluation using spread-plate method (SPM). *** *p* < 0.001.

**Figure 4 jfb-13-00035-f004:**
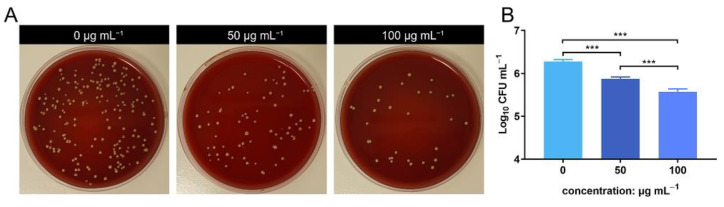
In vivo anti-bacteria activity of zinc oxide nanoparticles (ZnONPs) using the air pouch model infected with *Staphylococcus aureus* for 24 h. (**A**) Representative photos of the blood plate medium used for lavage harvest from the air pouch of groups treated with ZnONPs (0, 50, 100 μg mL^−1^). (**B**) Quantitative evaluation of the bacteria counts. *** *p* < 0.001.

**Figure 5 jfb-13-00035-f005:**
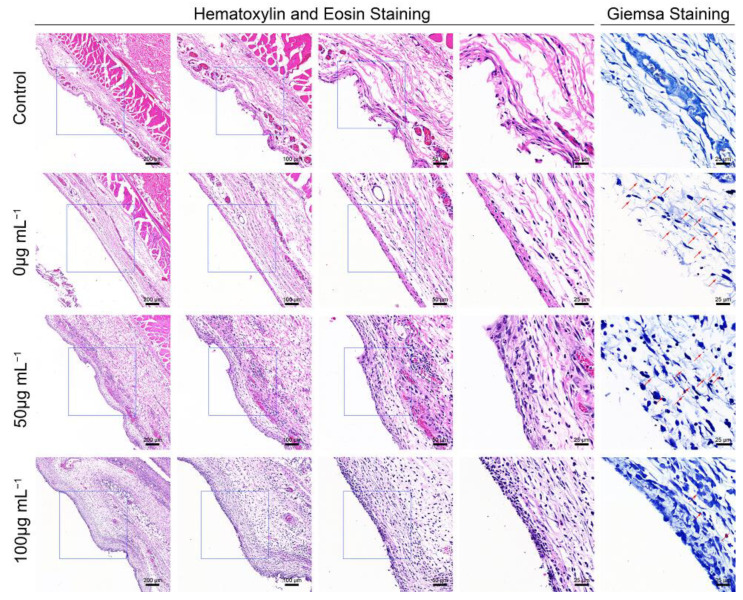
Histological evaluation of the full-thickness skin harvested from the air pouch after 24 h of the establishment of the model. Hematoxylin and eosin staining (50×, 100×, 200×, 400×) showed the neutrophil infiltration, and Giemsa staining (400×) mainly showed bacteria residual of the control group and infected groups treated with ZnONPs (0, 50, 100 μg mL^−1^). Red arrows indicated the bacteria residual.

**Figure 6 jfb-13-00035-f006:**
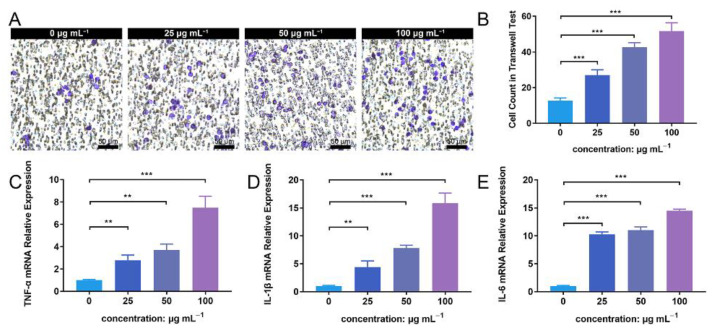
Cell migration and proinflammatory cytokine mRNA expression of polymorphonuclear neutrophils (PMNs) in the presence of bacteria treated with zinc oxide nanoparticles (ZnONPs). (**A**) Representative images of transwell tests demonstrating cell migration under the influence of *Staphylococcus aureus* and ZnONPs (0, 25, 50, 100 μg mL^−1^) for 2 h. (**B**) Quantitative calculation of migrant PMNs. (**C**–**E**) Proinflammatory cytokine mRNA expression (TNF-α, IL-1β and IL-6) of PMNs in co-culture with *Staphylococcus aureus* and ZnONPs (0, 25, 50, 100 μg mL^−1^) for 4 h. ** *p* < 0.01, *** *p* < 0.001.

**Figure 7 jfb-13-00035-f007:**
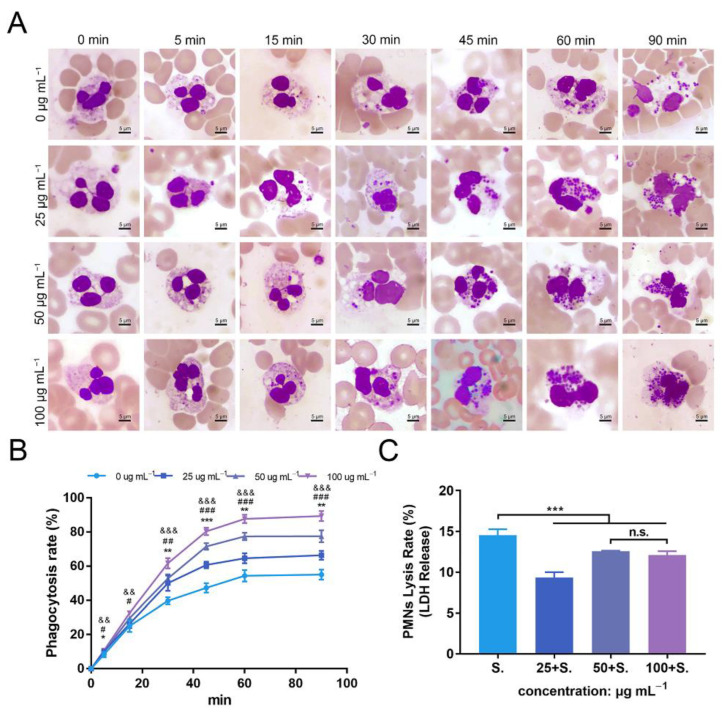
Phagocytosis of *Staphylococcus aureus* by polymorphonuclear neutrophils (PMNs) isolated from human whole blood, and PMN lysis rate, under co-culture with zinc oxide nanoparticles (ZnONPs). (**A**) Representative micrographs (1000×) of phagocytosis following Wright–Giemsa staining during continuous incubation (5, 15, 30, 45, 60, 90 min) with ZnONP (0, 25, 50, 100 μg mL^−1^) treatment. (**B**) Phagocytosis rate of *Staphylococcus aureus* by PMNs. * 25 μg mL^−1^ group versus 0 μg mL^−1^ group, & 50 μg mL^−1^ group versus 0 μg mL^−1^ group, #100 μg mL^−1^ group versus 0 μg mL^−1^ group; *, &, # *p* < 0.05; **, ##, && *p* < 0.01; ***, &&&, ### *p* < 0.001. (**C**) Cell lysis rate of PMNs after co-culturing with *Staphylococcus aureus* and ZnONPs (0, 25, 50, 100 μg mL^−1^). *** *p* < 0.001, n.s. represents no statistical significance.

**Figure 8 jfb-13-00035-f008:**
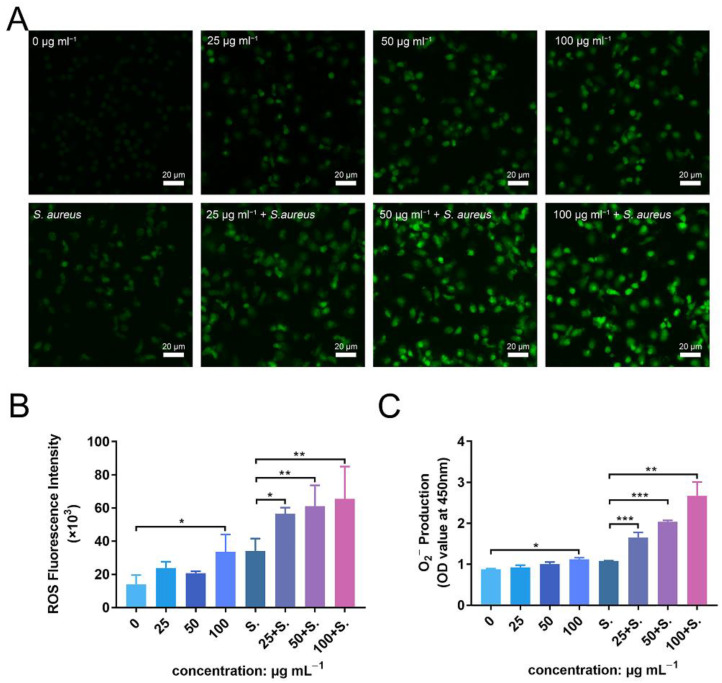
ROS and superoxide production by polymorphonuclear neutrophils (PMNs) after co-culturing with zinc oxide nanoparticles (ZnONPs), with or without bacteria. (**A**) Representative fluorescent images of ROS production in PMNs after 2 h of incubation with ZnONPs (0, 25, 50, 100 μg mL^−1^), with or without *Staphylococcus aureus*. (**B**,**C**) Quantitative fluorescence measurement of ROS and superoxide (O^2−^) production by PMNs after 2 h of incubation with ZnONPs (0, 25, 50, 100 μg mL^−1^), with or without *Staphylococcus aureus*. * *p* < 0.05, ** *p* < 0.01, *** *p* < 0.001.

## Data Availability

Not applicable.

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
