# Peer review of "Synergistic Antibacterial Effect of Zinc Oxide Nanoparticles and Polymorphonuclear Neutrophils"

_jfb, 2022, doi:10.3390/jfb13020035_

Round 1
Reviewer 1 Report
The topic of this study is of interest. However, some points must be addressed/clarified before publication.
Major concerns.
- Authors should provide more information on how PMNs were isolated. Simply citing that they were isolated by a neutrophil separating medium in M/M is simply insufficient. In addition, they do not refer to previous work.
- Authors should clarify why PMN phagocytosis capacity was determined from whole blood. It would be interesting to know if isolated PMNs would respond similarly or if the response would be different when they are in whole blood with other immune cells. This must be addressed especially knowing that the authors used isolated PMNs for studying other functions like ROS production and chemotaxis in vitro. This is not clear for me. What would be the effect if isolated human neutrophils were treated with ZnONPs and then with bacteria.
- Did the authors verify if ZnONPs could induce by themselves chemotactic migration of PMNs by simply adding the nanoparticles in the bottom well and PMNs in the upper chamber without the presence of bacteria in both chambers.
- The cytokine expression was determined by real-time PCR and it is not clear for me why the authors did not use ELISA to quantify them at the protein level. Because of that, authors should clearly mention that ZnONPs treatments increase the cytokine expression at the gene level and state that there is no proof that the cytokines are produced and released by PMNs at the protein level.
Minor concerns.
- The Y-axis tittle is missing in Figure 6, panel E.
- I suggest eliminating the last sentence of the conclusion. ‘‘However, this decision…’’.

Author Response
Point 1: Authors should provide more information on how PMNs were isolated. Simply citing that they were isolated by a neutrophil separating medium in M/M is simply insufficient. In addition, they do not refer to previous work.
Answer 1: Thanks for the suggestion. We have further added detailed methods of PMNs isolation and added a reference in the manuscript.
Point 2.1: Authors should clarify why PMN phagocytosis capacity was determined from whole blood. It would be interesting to know if isolated PMNs would respond similarly or if the response would be different when they are in whole blood with other immune cells. This must be addressed especially knowing that the authors used isolated PMNs for studying other functions like ROS production and chemotaxis in vitro. This is not clear for me.
Answer 2.1: The ex vivo bacteria-blood-ZnONP co-culture model was used to evaluate PMN phagocytosis. The reasons are as follows. Firstly, the aim was to best simulate in vivo conditions, this was for the consideration that bleeding is common around the infected site in early-stage infections after orthopedic surgery; Secondly, it was of great difficulty to smear the isolated PMNs sample equably in the glass slide methodologically, while it was far more feasible for the blood samples; Thirdly, we had mainly observed the behavior of PMNs in the whole blood sample, even though we did not use isolated PMNs.
Point 2.2: What would be the effect if isolated human neutrophils were treated with ZnONPs and then with bacteria.
Answer 2.2: Thanks for the question. We are sorry that we could not provide a clear answer for this. We would try to verify the effect in our further research.
Point 3: Did the authors verify if ZnONPs could induce by themselves chemotactic migration of PMNs by simply adding the nanoparticles in the bottom well and PMNs in the upper chamber without the presence of bacteria in both chambers.
Answer 3: Thanks for the question. We are sorry that we did not verify this. We believe that there would still be some PMNs migrated by simply adding the nanoparticles in the bottom well and PMNs in the upper chamber without the presence of bacteria in both chambers, but the number would not be as many as that when the presence of the bacteria in the bottom well.
Point 4: The cytokine expression was determined by real-time PCR and it is not clear for me why the authors did not use ELISA to quantify them at the protein level. Because of that, authors should clearly mention that ZnONPs treatments increase the cytokine expression at the gene level and state that there is no proof that the cytokines are produced and released by PMNs at the protein level.
Answer 4: Thanks for the suggestion. We have further emphasized in the manuscript that the cytokine expression was at the gene level but not at the protein level.
Point 5: The Y-axis title is missing in Figure 6, panel E.
Answer 5: We have corrected this error.
Point 6: I suggest eliminating the last sentence of the conclusion. ‘‘However, this decision…’’.
Answer 6: We have eliminated this sentence after careful consideration.
Reviewer 2 Report
The manuscript “Synergistic Antibacterial Effect of Zinc Oxide Nanoparticles and Polymorphonuclear Neutrophils” is devoted to investigation of ZnO NPs with cells representing innate immune system. The results show a cooperative effect of ZnO NPs and neutrophils against S. aureus. Here is an issue to be addressed:
The experiment of ROS and superoxide production in vitro includes the neutrophils incubation with various concentrations of ZnO NPs with or without bacteria. It was concluded that enhanced antibacterial effect elicited by neutrophils may be due to increased production of ROS and superoxide. ZnO NPs themselves may produce ROS; the mechanism of NPs toxicity is usually based on ROS generation. In this experiment the ROS generation by ZnO NPs during their incubation with bacteria (without neutrophils) is missed.
Author Response
Point: ZnO NPs themselves may produce ROS; the mechanism of NPs toxicity is usually based on ROS generation. In this experiment the ROS generation by ZnO NPs during their incubation with bacteria (without neutrophils) is missed.
Response:
Thanks for your comment.
It was reported that ZnO NPs themselves could produce massive ROS under visible light or UV exposure, and could barely produce ROS under dark conditions. In our experiment, the experiment was conducted in relatively dark conditions, including 2 hours' incubation. Thus, ZnO NPs may not significantly produce ROS.
As shown in Figure 2, the CCK-8 assay and LDH leakage assay had proved that ZnONPs in applied concentrations basically did not have toxicity on neutrophils, which meant that ZnONPs may not exert cell toxicity by producing ROS.
Considering the main purpose of this study, we did not set the group of "ZnO NPs + bacteria".
Reviewer 3 Report
The manuscript entitled “Synergistic Antibacterial Effect of Zinc Oxide Nanoparticles and Polymorphonuclear Neutrophils” by Kai Ye shows the potential effects and underlying mechanisms of ZnONPs on neutrophils to resist pathogen infection. Well written manuscript and carries considerable merits. While the work is of interest, after addressing the following comments, this manuscript is suitable for publication.
- Having a better introduction or comparing the results with other literature is suggested. Better introduction on synergic antibacterial effect and the employed materials suggested; recommended literature:
https://doi.org/10.1002/btm2.10254
- The quality of figure 1 is low. However, it may be due to the PDF conversion. Please check and use 600 DPI resolution. Also, font size should be in harmony in all figures.
- Text correction: Some typos and misspellings, e.g., “(1 × 105 per well)” should be corrected.
Author Response
Point 1: Having a better introduction or comparing the results with other literature is suggested. Better introduction on synergic antibacterial effect and the employed materials suggested; recommended literature: https://doi.org/10.1002/btm2.10254
Answer 1: The sentence"The nanocomposites of ZnO/Ag nanoparticles armed in chitosan/polyethylene oxide nanofibers have been successfully applied to exert synergic antibacterial effect with fibroblast cells[30]." has been added in the Introduction of our manuscript.
Point 2: The quality of figure 1 is low. However, it may be due to the PDF conversion. Please check and use 600 DPI resolution. Also, font size should be in harmony in all figures.
Answer 2: Yes, the PDF conversion decreased the figure quality. The original file of figure 1 was 300 DPI resolution and met adequate quality. In addition, we had tried our best to adjust the font size of each figure. However, because of the difference in editing software and figure size, it is quite difficult to achieve a completely uniform font size in all figures.
Point 3: Text correction: Some typos and misspellings, e.g., “(1 × 105 per well)” should be corrected.
Answer 3: We have corrected these errors.
Reviewer 4 Report
A very interesting and valuable manuscript for checking the effect of ZnONPs (nanomaterial with documented antibacterial activity) on the innate immune system, represented by polymorphonuclear leukocytes (PMNs), and determine whether they can act synergistically to resist pathogen infections.
In vitro results showed that ZnONPs exhibit understandable antibacterial activity at biocompatible concentrations but also enhance the antibacterial property of PMNs. In vivo experiments demonstrated the antibacterial effect of ZnONPs, accompanied by less infiltration of subcutaneous immune cells. Additionally, ex vivo and in vitro experiments revealed that ZnONPs enhanced the migration of PMNs, promoted their bacterial phagocytosis efficiency, proinflammatory cytokine (TNF-α, IL-1β, and IL-6) expression, and reactive oxygen species (ROS) production. So it has been shown, potential synergistic effects of ZnONPs on PMNs to resist pathogen infection. The results of these studies can be the basis for the design and preparation of materials containing ZnONPs in order to maximize their synergy with the innate immune system, thus promoting the host’s resistance to pathogen invasion.
There are numerous editorial errors in the work, e.g.
(1 × 106 CFU mL-1)
(0,25, 50, and 100 μg mL−1)
Figure 7. *25 μg mL-1 group versus 0 μg mL-1 group, &50 μg mL-1 group versus 0 μg mL-1 group, #100 μg mL-1 group versus 0 μg mL-1 group; *, &, # P < 0.05; **, ##, &&P < 0.01; ***, &&&, ### P < 0.001.
Staphylococcus aureus
References to literature have not been formatted in accordance with the journal's guidelines
Author 1, A.B.; Author 2, C.D. Title of the article. Abbreviated Journal Name Year, Volume, page range.
The missing part of the manuscript is the subsection Conclusions
Author Response
Answer: Thanks for the comments and suggestions. We have corrected these editorial errors, re-edited the references, and added conclusions in the manuscript.
Reviewer 5 Report
The authors of the manuscript :
"Synergistic Antibacterial Effect of Zinc Oxide Nanoparticles
and Polymorphonuclear Neutrophils"
did an excellent investigation into the antibacterial activity of ZnONP.
The work was done robustly and thoroughly by using different techniques to confirm the claims of the investigation.
There are few comments from my side:
1-The cytotoxicity/toxicity of ZnO NP should be mentioned from previous studies, if available
2- The authors should mention the future uses of ZnONP clearly, especially, where, in which combination and mention if the results of the animal model are expected to be as effective for further use in other models including human.
3-Many of the references are 2015 and below. The authors shall try to see newer publications
4-The reference style of mdpi is not applied. All the references need to be edited
5-Slight English grammar modifications needed. Some sentences are too long. But generally, the text is interesting and presents the results in excellent way. There is one mistake of writing CO2 instead of using CO2.
6-The interested reader will need few more examples in the introduction to be able to imagine the potential applications.
7-A short conclusion would be imperative to deliver the highlights and future perspectives of the investigations.
Otherwise, congratulations for the excellent work.
Good Luck.
Author Response
Point 1: The cytotoxicity/toxicity of ZnO NP should be mentioned from previous studies, if available
Answer 1: We have further added the sentence "The mechanism of cytotoxicity attributed to the dissolution of ZnONPs and the generation of ROS[20]." in the discussion.
Point 2: The authors should mention the future uses of ZnONP clearly, especially, where, in which combination and mention if the results of the animal model are expected to be as effective for further use in other models including human.
Answer 2: We have further added a paragraph about the future uses of ZnONP and the further use in other models in the discussion.
Point 3: Many of the references are 2015 and below. The authors shall try to see newer publications
Answer 3: Thanks for the suggestion. Many classical articles about ZnO NPs and neutrophils are early published, we would try to see newer publications in our future work.
Point 4: The reference style of mdpi is not applied. All the references need to be edited
Answer 4: We have re-edited the references.
Point 5: Slight English grammar modifications needed. Some sentences are too long. But generally, the text is interesting and presents the results in excellent way. There is one mistake of writing CO2 instead of using CO2.
Answer 5: Thanks for the comment. We have corrected mistakes including CO2.
Point 6: The interested reader will need few more examples in the introduction to be able to imagine the potential applications.
Answer 6: We have further added the example of the synergic antibacterial effect between ZnO NPs and fibroblast cells in the Introduction.
Point 7: A short conclusion would be imperative to deliver the highlights and future perspectives of the investigations.
Answer 7: We have further added the conclusion in the manuscript.